# A simple hydrogel device with flow-through channels to maintain dissipative non-equilibrium phenomena

Brigitta Dúzs [1] & István Szalai [1✉]

The development of autonomous chemical systems that could imitate the properties of living matter, is a challenging problem at the meeting point of materials science and nonequilibrium chemistry. Here we design a multi-channel gel reactor in which out-of-equilibrium conditions are maintained by antagonistic chemical gradients. Our device is a rectangular hydrogel with two or more channels for the flows of separated reactants, which diffuse into the gel to react. The relative position of the channels acts as geometric control parameters, while the concentrations of the chemicals in the channels and the variable composition of the hydrogel, which affects the diffusivity of the chemicals, can be used as chemical control parameters. This flexibility allows finding easily the optimal conditions for the development of nonequilibrium phenomena. We demonstrate this straightforward operation by generating diverse spatiotemporal patterns in different chemical reactions. The use of additional channels can create interacting reaction zones.

[1] Institute of Chemistry, Eötvös Loránd University, Budapest, Hungary. ✉email: szalai.istvan@chem.elte.hu

Living systems continuously consume energy, which is often provided by chemical fuels, to stay in dissipative non-equilibrium states, to maintain spatiotemporal control over the biochemical processes, and to sustain their structure and function. Synthetic systems that can mimic some of these features are not only theoretically interesting but may provide a unique possibility to create a new type of material and autonomous chemical devices. For instance, non-equilibrium supramolecular self-assembly offers a way to make self-healing or highly adaptive materials[1,2]. In the same spirit, stimuli-responsive polymer gels are used to build sensors and actuators, which can be applied in various fields, such as electronics, biomedical applications, or soft robotics[3]. Flow reactors are capable of keeping a reacting system far from equilibrium, but to have a spatiotemporal control over the operation of the system, e.g., to synthesize the desired supramolecular structures or drive the motion of a responsive polymer, is a challenging problem[1,2]. To create a functioning non-equilibrium device, the use of oscillatory or bistable chemical reactions is a promising direction. Periodic supramolecular systems have been made by coupling chemical oscillators with the aggregation of functionalized gold nanoparticles[4], by a micelle-to-vesicle transition of oleic acid-based surfactants[5], and by an assembly of a polyethylene glycol-functionalized polymer[6]. The motion of a responsive hydrogel can be also driven in the same way, but in this case, the diffusive transport of the chemicals into the gel is a key issue[7–9]. Typically, the gel is immersed into a constant or oscillatory chemical environment and the reagents diffuse from outside into the gel, where they react and induce periodic changes[7–9]. This design, where the fresh reagents are supplied by diffusion from the outer solution, may cause some limitations in the potential practical applications. We propose to turn this configuration inside out by using a hydrogel with flow-through channels to create a spatiotemporal non-equilibrium system. This simple reactor provides a way to perform a chemical reaction at out-of-equilibrium conditions or even to create a chemically fueled device. The dynamics of reaction-diffusion (RD) systems at the boundary conditions relevant in this design, even if the gel is non-responsive, is unexplored.

We present experimental evidence for the formation of various types of RD phenomena in a reactor with the following attributes: (i) fluidic channels provide the continuous supply of the counter-diffusing chemicals, and (ii) a hydrogel serves as the non-convective reaction zone for pattern formation (Fig. 1a). The reactor geometry is characterized by two parameters (Fig. 1b), the thickness of the cover domain ($h$), and the distance between the channels ($w$). Both can be tuned according to the properties of the applied reaction. This configuration generates cross-gradients of the reagents, which often serves as the originator of natural symmetry breaking processes, like the formation of stripes with sharp boundaries in the Wolpert concept on biological pattern formation[10,11]. To maintain fixed concentration gradients, the so-called two-side-fed gel reactor (TSFR), where two well-stirred tanks are separated by a hydrogel, has already been successfully used to produce various non-equilibrium phenomena, like chemical waves and Turing patterns[10,12], standing structures in synthetic biochemical systems[13], and it has also been used for the synthetic purpose to control the formation of a fibrous hydrogel network[14]. Our design reconsiders the TSFR concepts: instead of well-stirred reservoirs, we use fluidic channels embedded in the hydrogel to provide continuous sources of the separated reactants. This simple setup offers the necessary flexibility and variability for the potential applications. A video that presents the reactor and the gel preparation process is available as Supplementary Movie 1.

## Results

**Stationary stripe formation**. As the first step, we used a two-channel setup to create a stationary stripe standing perpendicular to the direction of the concentration gradients, that is maintained by the different compositions in the two flows. The necessary positive feedback process is either provided by a homogeneous autocatalytic reaction, i.e., $H^+$ autoactivated oxidation of sulfite ions with bromate ions[15] (Fig. 2a) or by a self-activated heterogeneous process, i.e., precipitate formation between calcium and carbonate ions (Fig. 2b–e). Although the chemistry of these reactions is quite different, the generated three-zoned patterns have the same macroscopic structure as these are built up from a high extent of reaction zone bounded with two low extents of reaction zones with different chemical compositions. In the case of the $H^+$ autoactivated reaction, we used a reactor configuration with $w = 2.5$ mm, since the relatively long time scale of this reaction requires a steep gradient (short time scale diffusive feed). The appearance of the sharp acidic stripe from a non-structured initial state was initiated by increasing the gradient of $[H^+]$ (Fig. 2a). The precipitate band was created in a reactor with $w =$

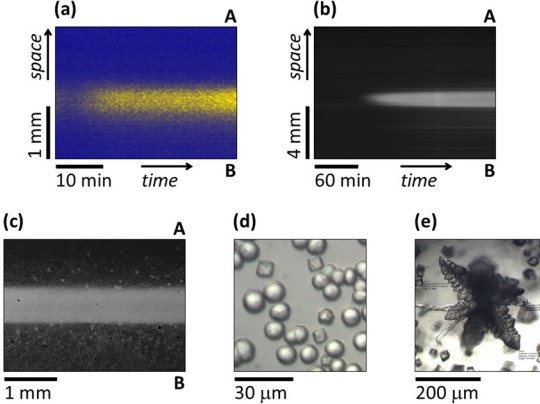

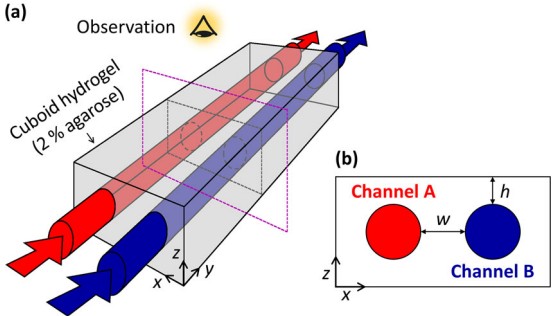

**Fig. 1 Scheme of the two-channel open gel reactor.** The three-dimensional orientation of the setup (**a**). Characteristic dimensions of the reactor in the x–z cross-section (**b**). The agarose gel body is fed from two liquid channels with different content (indicated by red and blue). The distance of the channels ($w$) and the thickness of the top and bottom gel layers ($h$) are adjustable parameters.

**Fig. 2 Three-zoned standing reaction-diffusion structures in two-channel experiments.** Acidic stripe formation in the $H^+$ autocatalytic bromate-sulfite reaction (**a**) and precipitate formation in the $CaCO_3$ system (**b–e**). Space-time plots of the stabilization of the patterns (**a**, **b**), snapshot in the x–y plane at 4 h (**c**), and optical microscope images made right after the synthesis (**d**, **e**). The pictures are taken from the reacting zone between the channels denoted by A and B. Experimental conditions: **a** $[BrO_3^-]_A = 200$ mM, $[H_2SO_4]_B = (5{\rightarrow})6$ mM, $[SO_3^{2-}]_{A,B} = 80$ mM, $[Fe(CN)_6^{4-}]_{A,B} = 10$ mM, $[BCG]_{A,B} = 0.1$ mM, $w = 2.5$ mm, $T = 35\,°C$; **b–e** $[Ca^{2+}]_A = [CO_3^{2-}]_B = 150$ mM, $w = 10$ mm, $T = 25\,°C$.

10 mm providing shallow gradients (long time scale diffusive feed), that fit the short time scale of the precipitation process. The gel was empty at the beginning of the experiments and the concentration gradients of calcium and carbonate ions changed gradually until the precipitate band reached its final width (Fig. 2b and Supplementary Movie 2). The final sharp precipitate zone (Fig. 2c) contains mainly spherical and cubic (Fig. 2d) and some flower-shaped (Fig. 2e) crystals. The spherical and flower form is typical for the metastable vaterite, while the cubic is typical for calcite. In both cases, the continuous supply of the reagents provides non-equilibrium conditions and therefore maintains the three-zoned patterns.

**Dynamical structures**. To create a dynamical structure, e.g., spatiotemporal waves, two homogeneous chemical oscillators were chosen: the bromate-sulfite-ferrocyanide (BSF)[15] and the chlorite-iodide-malonic acid (CIMA)[16] systems. These are essentially different types of chemical oscillators, but both are known to produce RD patterns[12,17]. In the BSF reaction, the positive feedback is provided by the $H^+$ autocatalytic reaction of the initial chemicals, so the continuous supply of those is necessary for the time-periodic behavior. The CIMA reaction can show sustained oscillations for a while even without the external supply of the reagents because the key species of the dynamics are produced from their precursors, which are the initial reagents. In the CIMA system, the positive feedback is ensured by substrate inhibition, as an iodide consuming reaction that slows down at higher concentrations of iodide. These experiments were made in a reactor with $w = 2.5$ mm. Bromocresol green pH indicator was used in the BSF system and poly(vinyl alcohol) (PVA) in the CIMA system to visualize the pH and $I^-$ structures, respectively. The state of the moving structures was reached either from a spatial stationary state by changing the gradient of $[H^+]$ in the BSF and that of the malonic acid (MA) in the CIMA case. In this reactor, the dynamic instability that leads to the formation of waves is localized in a narrow zone (Fig. 3a, b). The position of this zone can be slightly tuned by the concentrations in the channels, but under the applied conditions the dynamical gel zone is located almost in the middle in the BSF system (Supplementary Movie 3) and shifted towards channel B (that consists of $I^-$ and MA) in the CIMA system (Supplementary Movie 4). The period of the pH waves is about 6 min (Fig. 3a) while that of the $I^-$ waves is only 14 s (Fig. 3b).

The forming sustained spatiotemporal waves make the BSF reaction a good model system to test the adaptability of our reactor concept. Four different constructions were used with different sizes (Supplementary Fig. 1). At first, we tested a miniaturized design with smaller channels ($d = 1.3$ mm) but with the same distance between the channels ($w = 2.5$ mm) and verified the existence of pH waves (Supplementary Fig. 2a). Then polyacrylamide hydrogel was used instead of agarose to demonstrate the applicability of more elastic and less brittle material by using the original reactor ($d = 7.0$ mm, $w = 2.5$ mm). In this case, we could also detect pH waves (Supplementary Fig. 2b), in spite of the technical problems caused by the initial swelling of the synthesized polyacrylamide gel (see "Experimental methods" section). However, the modification of the gel matrix affects the dynamics as well. Polyacrylamide gels always contain free carboxyl groups, which act as reversible $H^+$ binding sites in the gel, causing long-range inhibition in an $H^+$ activated RD system and leading to stationary Turing structures[18]. In our experiments, a line of acidic spots appeared in the middle zone when polyacrylamide gel was used (Supplementary Fig. 2c, d).

Dynamic RD structures can also be created by using a cation-anion pair, which forms a precipitate that can dissolve in excess of

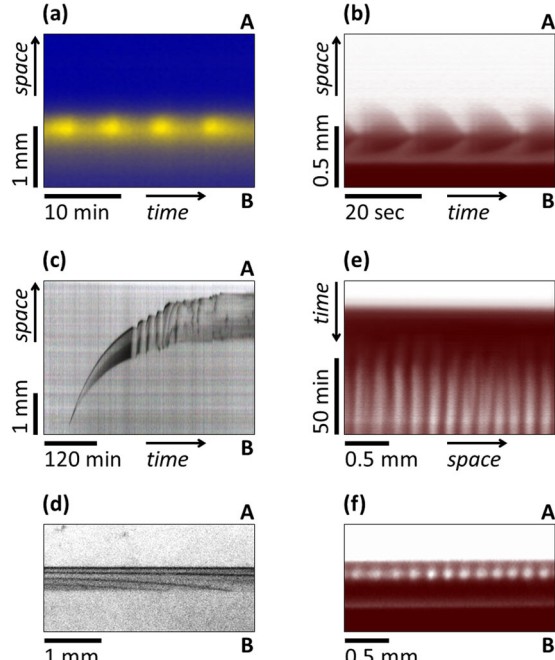

**Fig. 3 Moving reaction-diffusion phenomena and standing spots in the two-channel setup.** Waves (**a–d**) and localized Turing patterns (**e–f**) in a pH oscillator (**a**), in a chlorite oscillator (**b**, **e**, **f**) and in a precipitation-redissolution system (**c**, **d**). Space-time plots (**a–c**, **e**) of the periodic phenomena and snapshots made at 4 h (**d**) and at 90 min (**f**). The pictures are taken from the reacting zone between the channels denoted by A and B. Experimental conditions: **a** $[BrO_3^-]_A = 200$ mM, $[H_2SO_4]_B = 9$ mM, $[SO_3^{2-}]_{A,B} = 80$ mM, $[Fe(CN)_6^{4-}]_{A,B} = 20$ mM, $[BCG]_{A,B} = 0.1$ mM, $w = 2.5$ mm, $T = 35$ °C; **b** $[ClO_2^-]_A = 20$ mM, $[IO_3^-]_A = 1.5$ mM, $[NaOH]_A = 12$ mM, $[I^-]_B = 2.5$ mM, $[MA]_B = 16$ mM, $[H_2SO_4]_B = 10$ mM, $[PVA]_{A,B} = 1.5$ g L$^{-1}$, $w = 2.5$ mm, $T = 5$ °C; **c**, **d** $[Al^{3+}]_A = [H_2SO_4]_A = 0.1$ M, $[OH^-]_B = 2.5$ M, $w = 10$ mm, $T = 25$ °C; **e**, **f** $[ClO_2^-]_A = 10$ mM, $[IO_3^-]_A = 2$ mM, $[NaOH]_A = 12$ mM, $[I^-]_B = 2$ mM, $[MA]_B = 6.5$ mM, $[H_2SO_4]_B = 10$ mM, $[PVA]_{A,B} = 1.5$ g L$^{-1}$, $w = 2.5$ mm, $T = 5$ °C.

one of the forming ions[19]. We used a flow-through reactor with $w = 10$ mm and fed the channels with acidic $AlCl_3$ and NaOH solutions (Fig. 3c, d). The applied larger $w$ was necessary to maintain the non-equilibrium conditions long enough for the generation of quasi-sustained spatiotemporal oscillations. The gel was empty at the beginning of the experiments and in the presented example, a thin precipitate band appeared after 1 h and became thicker and structured as it propagated towards channel A (that consists of acidic $AlCl_3$) (Fig. 3c). Two hours later, when the position and the thickness of the band were nearly stabilized, spatiotemporal waves developed with a period of 20 min (Fig. 3d). The waves spread through the whole 0.85-mm-thick precipitate band and formed spirals for 2 h (Supplementary Movie 5). After this long-lasting transient period, the pattern reached its steady-state close to channel A. The formation and dissolution of precipitate filaments were observed for more than 12 h.

In the next series of experiments, we created sustained stationary Turing patterns in a two-channel flow-through gel reactor. We used the CIMA reaction, which is well-known to generate Turing patterns in the presence of a supercritical amount of PVA in a TSFR[12]. The cross-gradients of the reactants between the channels determine a narrow zone where the conditions of Turing instability are fulfilled, as the diffusivity of $I^-$ in the agarose is appropriately tuned by adding PVA into the gel. In that region highly ordered line or lines of spots appear with a

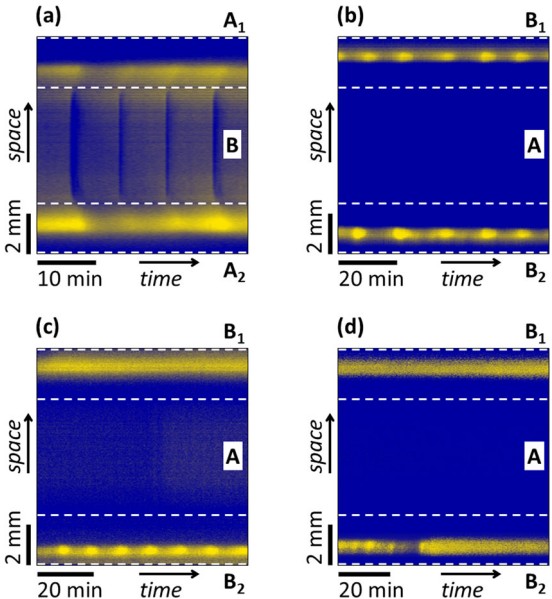

**Fig. 4 Development and interaction of two reaction zones in different types of symmetric three-channel configurations in the BSF reaction represented by space-time plots.** The pictures show these zones, separated by dashed lines from the region of the middle channel. A and B denote channels containing $H_2SO_4$ and oxidant, respectively, while the indexes refer to channels of the same reagents but with different concentrations. Experimental conditions: **a** $[BrO_3^-]_{A1,A2} = 200$ mM, $[H_2SO_4]_B = 9$ mM, $[Fe(CN)_6^{4-}]_{A1,B,A2} = 10$ mM; **b**–**d** $[BrO_3^-]_A = 200$ mM, $[Fe(CN)_6^{4-}]_{B1,A,B2} = 20$ mM, **b** $[H_2SO_4]_{B1,B2} = 11$ mM, **c** $[H_2SO_4]_{B1} = 9$ mM, $[H_2SO_4]_{B2} = 13$ mM, **d** $[H_2SO_4]_{B1} = 8$ mM, $[H_2SO_4]_{B2} = 11$ mM; $[SO_3^{2-}]_{A1,B,A2,B1,A,B2} = 80$ mM, $[BCG]_{A1,B,A2,B1,A,B2} = 0.1$ mM, $w = 2.5$ mm, $T = 35$ °C.

wavelength of 0.25 mm (Fig. 3f). In our experiments, the pattern typically reached its final form within 20 min (Fig. 3e). More complex dynamics, e.g., the interaction of waves and patterns were also observed (Supplementary Movie 6).

The use of three flow-through channels allows us to test the appearance of diffusive coupling between spatially separated RD zones. We used equidistant, parallel channels that define two RD zones with the same $w$. Since we have two different mixtures (A and B) and three channels, two distinct configurations can be made, A-B-A and B-A-B. In the case of the BSF reaction, the reagent separation is the following: A contains the oxidant ($BrO_3^-$) and B contains the acid ($H_2SO_4$) while the other chemicals were supplied in both channels. In the oxidant-acid-oxidant configuration, we observed interaction between the two RD domains as spatiotemporal waves developed not only in the gel zones between the channels but also in the cover gel domain above the middle channel (Fig. 4a and Supplementary Movie 7). Contrary, in the acid-oxidant-acid configuration, we have not found evidence of such coupling, as the phases of the waves in the two RD zones were not synchronized (Fig. 4b and Supplementary Movie 8). But, in this setup when the acid concentration in the side channels differs, we observed another type of communication. We set an acid concentration in $B_1$ and $B_2$ to a value ($[H_2SO_4]_{B1,B2} = 9$ mM) that is just below the value necessary to get spatiotemporal oscillations, therefore, two stationary acidic zones were formed, one between channels $B_1$ and A and the other one between A and $B_2$. In the $B_2$ channel, the acid concentration was increased step-by-step starting from the same value, while the other was kept at the original value. Above a critical value of $[H_2SO_4]_{B2}$, waves appeared between channels A and $B_2$, but the acidic stripe between channels $B_1$ and A remained stationary

(Fig. 4c and Supplementary Movie 9). Even a further increase of $[H_2SO_4]_{B2}$ until 30 mM has not induced periodic behavior between channels $B_1$ and A. However, a decrease of $[H_2SO_4]_{B1}$ to 8 mM has stopped the spatiotemporal oscillations between channels A and $B_2$ and stabilized a stationary stripe state (Fig. 4d). Thus a change in the concentration of the acid on one side can stop the oscillations on the other side.

**Numerical simulations**. To clarify the effects of the reactor geometry on the dynamics numerical simulations were made by the Rábai model of pH oscillators[20], which represent the core mechanism of the BSF reaction (for the details of the model see Supplementary Methods)[16]. This model (Fig. 5a) includes a positive feedback process, that is the $H^+$ autocatalytic oxidation of a weak acid (HA), and negative feedback, that is the $H^+$ consuming oxidation of component (C). The cross-section of the channels was approximated by square domains. In the left channel (L) the mixture of the oxidant (B), the unprotonated form of the weak acid ($A^-$) and C, while in the right channel (R) the mixture of $A^-$, C, and $H^+$ flows (Fig. 5b). Here we applied a high enough concentration of C to promote the development of an oscillatory instability. Depending on the concentration of $H^+$ in the flow ($[H^+]_R$) and the geometry ($w$ and $h$), three types of spatial stationary states, denoted by F, M1, and M2 (Fig. 5c–e) and spatiotemporal oscillations can form (Fig. 5f, g). Contrary to the unstructured F state, at both M states a sharp acidic zone develops in the middle top region of the gel. In this situation, the acidic zone develops from the middle top zone, but not at the center due to the asymmetric conditions. The M2 state, where the acidic zone appears in between the channels, is favored at higher values of $[H^+]_R$ and $w$. The pH waves always start from the top domain and propagate in the direction of the $z$-axis. Type 1 waves are localized into the top domain (Fig. 5f) and type 2 waves develop along the entire zone between the channels (Fig. 5g). At a fixed $w$, the domain of type 2 oscillations in the $h$-$[H^+]_R$ parameter plane significantly widens as $h$ increases above $h = 0.5$ (Fig. 5h). Above $h = 1.0$ the oscillations become localized at lower values of $[H^+]_R$ and the domain of type 2 waves does not change significantly as $w$ increases. At a fixed $h$, type 2 waves appear only above the critical $w$, which exceeds the value of $h$ (Fig. 5i). As $w$ increases, the width of the oscillatory domain does not change significantly in the $w$-$[H^+]_R$ parameter plane but type 2 waves become favored at the expense of type 1 waves. We found a smooth transition between the two types of waves, therefore the border-line of these behaviors is indicated by the dotted line in Fig. 5h, i.

According to these phase diagrams, the role of the geometric control parameters is more to set the available states. For instance, the dynamics are simpler at low values of $h$ or at high values of $w$, as only F, M2, and type 2 waves appear, but become richer at the opposite cases. An appropriate chemical control parameter, like $[H^+]_R$ here, allows inducing transitions between the available states. We anticipate, that these results are not specific for pH oscillators, and the various families of nonlinear reactions, which show oscillations only in presence of continuous supplement of the reagents[21], would show similar dynamics.

**Discussion**

The presented simple and flexible reactor design can maintain RD systems far from equilibrium and create a self-organized spatio-temporal state in a hydrogel. However, this practical simplicity goes hand in hand with some theoretical difficulties. From a theoretical point of view, the boundary conditions are more complex than in the typical open reactors, which are often used to study RD phenomena[18]. The formation of patterns is not trivial

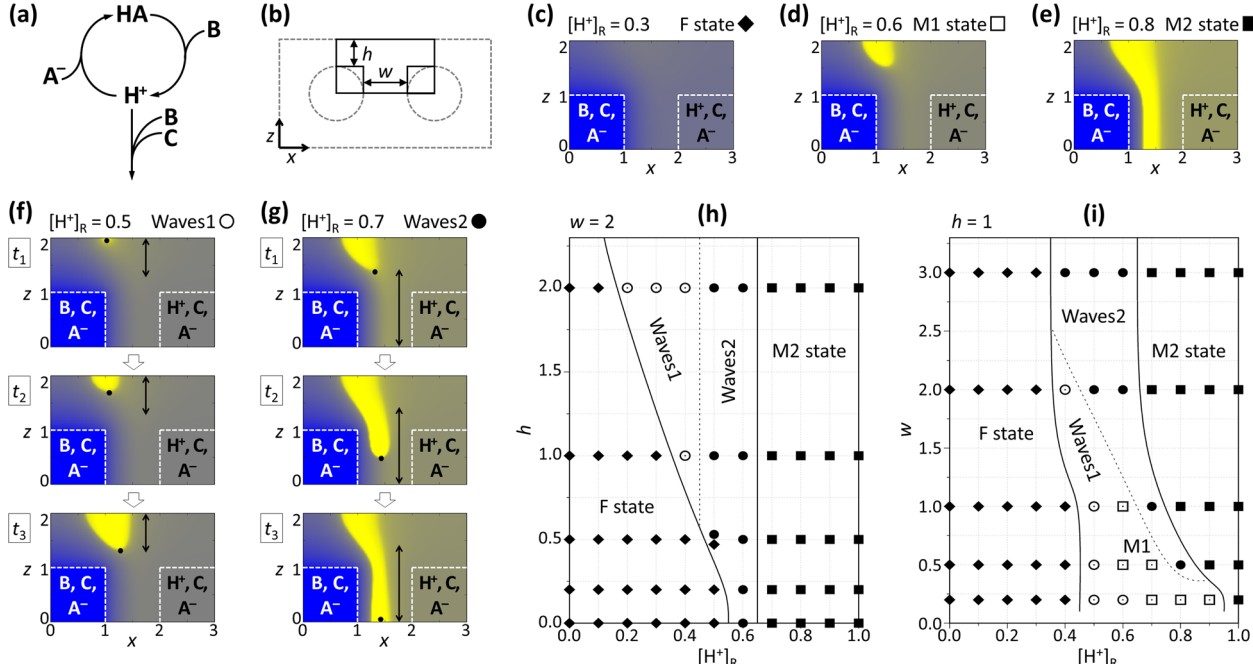

**Fig. 5 Simulated two-dimensional spatiotemporal phenomena in the two-channel configuration for pH oscillators.** Sketch of the core mechanism of the Rábai model (**a**). Concept of the geometrical approximation used in the simulations (**b**). Snapshots of F (filled diamond), M1 (open square), and M2 (filled square) stationary spatial states (**c–e**) and consecutive snapshots of two types of spatiotemporal oscillations (open circle and filled circle) (**f, g**). The positions of the channels are indicated by dashed lines and with the chemical compositions. Here $[H^+]_R$ indicates the acid concentration in the Right channel. Phase diagrams of the dynamics in the $h$-$[H^+]_R$ (**h**) and $w$-$[H^+]_R$ (**i**) parameter planes. For further details, see Supplementary Methods.

in this flow-through design, but as we have shown, the aligned optimization of the geometrical and chemical parameters works well. In practical applications, the relatively low consumption of the input chemicals and the single-body design are additional profitable properties. Although it was not our intention, these experimental results also demonstrated the robustness of the applied chemical systems in producing spatiotemporal patterns. The experimental observation of long-lasting dynamic precipitate patterns and diffusively communicating RD zones are the first achievements of this new design. The thickness of the diffusive zone can be easily tuned by the distance of the channels, and the miniaturization of the reactor was also achieved by decreasing the channel diameter. Based on these features, we anticipate that flow-through gel reactors can be effectively used to explore the dynamics of a wide range of chemical and biochemical reactions and also to combine them with responsive materials. The study of the behavior of supramolecular self-assembly under conditions of continuous supply is another challenging problem, where these reactors can be potentially applied[22]. By creating an array of channels in a hydrogel with different geometric arrangements would open new possibilities to create complex geometrical constraints on pattern formation in an open system.

## Methods

**Experimental methods.** The experiments were carried out in a 20 mm (in case of the normal-sized $w = 2.5$ mm reactor) or 30 mm (in case of $w = 10$ mm and in the three-channel setup) or 7.5 mm (in case of the miniaturized $w = 2.5$ mm reactor) ($x$) × 40 mm ($y$) × 10 mm ($z$) cuboid piece of 2 m/V% agarose hydrogel (Sigma-Aldrich, A0169) or polyacrylamide hydrogel. The casting of the gel was performed in a home-made rectangular plexi mold where the channels were created by plexi tubes (diameter 4.5 mm) or by stainless steel needles (diameter 1.3 mm). Preparation of the agarose gel body: after pouring the hot agarose into the mold it was covered by a plexi plate. Once the gel has solidified the plexi tubes were pulled out from the gel, but not completely, as these tubes were used as connectors for the input and output flows. The distance between the closest points of the channels ($w$) was 2.5 mm in the case of BSF and CIMA reactions and 10 mm in the case of CaCO$_3$ and Al(OH)$_3$/[Al(OH)$_4$]$^-$ systems. The thickness of the gel layers below

and above the channels was $h = 2.0$ mm except in the miniaturized reactor where it was $h = 1.2$ mm. The polyacrylamide gel was prepared as follows: acrylamide (AA, Fluka, ≥99%), $N,N'$-methylene-bisacrylamide (BAA, Sigma, ≥98%), and bromo-cresol green indicator (BCG, Sigma-Aldrich, 90%, p$K_a = 4.8$) were dissolved in 4.8 mL distilled water. Then 0.6 mL ammonium peroxodisulfate (APS, Sigma) and 0.6 mL triethanolamine (TEA, Sigma, ≥99%) solutions were added dropwise, with vigorously mixing for 1–2 min. The final concentrations were [AA] = 2.8 M, [BAA] = 13 mM, [BCG] = 0.1 mM, [APS] = 13 mM, [TEA] = 30 mM. The mixture was poured into the plexi mold and was covered with a plexi plate for 24 h. Then the channels were pulled out, the gel body was removed from the mold and the channels were washed with 0.1 mM BCG solution for 48 h. Due to the washing, the whole gel body swelled but the distance between the channels remained the same ($w = 2.5$ mm). After that, junction tubes (diameter 7.0 mm) were inserted into the channels, the polyacrylamide gel was placed into a bigger mold, was poured around with 2 m/V% hot agarose containing 0.1 mM BCG, and was covered with a plexi plate. The continuous inflows of the reactants were maintained by peristaltic pumps. The feed solutions of the initial chemicals were stored in separate reservoirs but entered premixed into the channels. The reactants were distributed as follows: BSF system: Reservoir 1: NaBrO$_3$ (Sigma-Aldrich, ≥99%), Reservoir 2: Na$_2$SO$_3$ (Sigma-Aldrich, puriss), Na$_4$[Fe(CN)$_6$]·10H$_2$O (Sigma-Aldrich) and BCG, Reservoir 3: H$_2$SO$_4$ (diluted from 2.5 mol L$^{-1}$ standard solution, VWR); Channel A is fed from Reservoir 1 and 2 while Channel B is fed from Reservoir 2 and 3. CIMA system: Reservoir 1: NaClO$_2$ (Fluka, ≥80%), KIO$_3$ (Sigma-Aldrich, ≥98%), NaOH (VWR) and poly(vinyl alcohol) (PVA, Aldrich, 80% hydrolyzed, $M_w = 9000$–10,000), Reservoir 2: KI (VWR, ≥99%), H$_2$SO$_4$ (diluted from 1.0 mol L$^{-1}$ standard solution, VWR) and PVA, Reservoir 3: malonic acid (Fluka, ≥98%), Reservoir 4: ion-exchanged water; Channel A is fed from Reservoir 1 while Channel B is fed from Reservoir 2, 3, and 4. Concentrations of BCG and PVA indicators were set in the melted agarose and were maintained by the flows. CaCO$_3$ precipitation system: Reservoir 1: Ca(NO$_3$)$_2$·4H$_2$O (Sigma-Aldrich), Reservoir 2: Na$_2$CO$_3$ (Reanal); Channel A is fed from Reservoir 1 while Channel B is fed from Reservoir 2. Al(OH)$_3$/[Al(OH)$_4$]$^-$ system: Reservoir 1: AlCl$_3$·6H$_2$O (Spectrum-3D) and H$_2$SO$_4$ (diluted from 1.0 mol L$^{-1}$ standard solution, VWR), Reservoir 2: H$_2$SO$_4$, Reservoir 3: NaOH (VWR), Reservoir 4: ion-exchanged water; Channel A is fed from Reservoir 1 and 2 while Channel B is fed from Reservoir 3 and 4. All solutions were prepared daily with ion-exchanged water and the chemicals were used without further purification. The flow rate in each channel was 200 mL h$^{-1}$ in the case of BSF and CIMA systems and 100 mL h$^{-1}$ in the case of CaCO$_3$ and Al(OH)$_3$/[Al(OH)$_4$]$^-$ systems. The experiments were performed at 35 °C in the case of BSF system, at 5 °C in the case of CIMA system, and at 25 °C in all other cases. The input feed concentrations of the reagents ([X]$_{Channel}$) are indicated in the text. The reactor was enlightened from one side by a flat LED source while the camera was fixed on the other side. The pictures were taken by an AVT Stingray F-033B

(14 bit) camera and were recorded by the Streampix (Norpix) software. We applied a 520 nm bandpass filter in BSF and CIMA experiments to enhance contrast and to protect ferrocyanide from light. We used the ImageJ program for image processing.

**Numerical methods**. The numerical simulations were made by using the five-variable Rábai model of pH oscillators[20]. The equations are shown in Supplementary Methods. The partial differential equations were discretized with a standard second-order finite difference scheme on a $200 \times 100$ mesh. The resulting systems were solved by the SUNDIALS CVODE solver using a backward differentiation formula method[23]. The absolute and relative error tolerances were $10^{-12}$ and $10^{-7}$, respectively.

## Data availability
The data that support the findings of this study are available from the corresponding author upon reasonable request.

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

## Acknowledgements
We thank the support of the ÚNKP-19-3 New National Excellence Program of the Ministry for Innovation and Technology and the support of the National Research, Development and Innovation Fund (119360). This work was completed in the ELTE Thematic Excellence Program (Szint+) and Institutional Excellence Program (1783-3/2018/FEKUTSRAT). We also thank the help of Mr. Dániel Papp in the preparation of Supplementary Movie 1.

## Author contributions
B.D. performed the experiments and I.S. supervised the project. Both the authors contributed to the numerical simulations and wrote the manuscript.

## Competing interests
The authors declare no competing interests.
