## [Peer Review File · Communications Chemistry]

Reviewers' comments:

Reviewer #1 (Remarks to the Author):

The paper by Duzs and Szalai addresses an important issue of designing a feeding system for open chemical systems that are kept out-of-equilibrium by a continuous supply of reagents. The authors demonstrated how to maintain out-of-equilibrium states by the flow of reagents through the channels in a piece of agarose hydrogel. Although this idea is relatively straightforward, I have not seen its realization in the literature. The method is broadly applicable for many chemistries and functions; thus, it will be interesting to many researchers working in the fields of systems chemistry and out-of-equilibrium materials. The paper is reasonably clearly written.

Nevertheless, the focus of the paper is shifted from the design of the flow-through gel reactors. Only two examples (two and three-channel configurations) are shown. Very limited discussion of the assembly of the reactor exists in the main text. Only a brief description of the construction of the reactor is given in the methods section. Actually, the design of the reactor should be the most instructive part of the paper shown in enough detail that the community can use it for various tasks. In this regard, I suggest the following three additions: (i) to provide a detailed description of the reactor design including the construction of the mold, types of the tubes, photos of different stages of the assembly process; (ii) to show designs with smaller channels (e.g. 1 mm in diameter) because the miniaturization of the setup will significantly broaden its applicability, especially to biochemical systems; (iii) to demonstrate a setup from polyacrylamide hydrogel because polyacrylamide hydrogels are much more elastic and less brittle than agarose hydrogels making them more applicable for designing functional materials.

Finally, there are a few small issues that should be addressed:

p.1 , the term "communicating reaction zones" is unclear in the context of the abstract without reading the full text of the manuscript.

A figure for the three-channel setup should be given.

p.4 Regarding the following sentence "In both cases, the continuous supply of reagents maintains non-equilibrium conditions and stabilizes the three-zoned patterns". It should be considered that in contrast to [H⁺] pattern, CaCO₃ pattern is not stationary because CaCO₃ crystals grow constantly.

Reviewer #2 (Remarks to the Author):

In this paper, a device is constructed with liquid channels embedded in a gel matrix and reactions taking place by diffusion of species between channels. This is an interesting idea that is well executed with some relatively simple examples. The paper has the possibility to influence developments in the field. There might be an opportunity to further control the diffusive properties of selected species by choice of gel components, for example, or macromolecules through the gel pore size in order to form non-equilibrium structures. Another idea could be to trap cellular objects like giant vesicles in the central channel then continuously feed different reactants in the 2 other channels. The manuscript is clearly written and the figures are well presented. I have only minor comments:

To be clear, in figure 2, the space shown is that between the two channels, w , and the crystals shown in 2d/e are formed in the gel matrix, not the liquid channel?

Is it possible to take images from the side also, to view the 3 dimensional structure of the Turing patterns?

Discussion section could be renamed conclusions.

Reviewer #3 (Remarks to the Author):

Summary:

The authors proposed an interesting reaction-diffusion (RD) system which allows users to maintain non-equilibrium conditions and form different spatiotemporal patterns.

The RD system is created by the addition two or more parallel flow channels within a rectangle hydrogel. The two channels would then form antagonistic chemical gradients across the hydrogel in which chemicals react. The authors first demonstrated the RD system can form stable reaction conditions by the CaCO_3 precipitation reaction, creating stable three-zoned patterns. The authors then showed that they could also form stable spatiotemporal patterns by creating pulse-like waves traveling within the gel with the H^+ autocatalytic bromate-sulfite reaction. Moreover, the authors go on and couple multiple chemistry into the RD system and created more spatiotemporal patterns such as Turing patterns, showing that the wide application that the RD system can be coupled with.

Another strength of the RD system is the flexibility and ability to run parallel experiments. By tuning the concentrations of the inlet chemicals into the flow channels, and also the relative distance between the channels, the authors demonstrated the chemical and geometric knobs that are able to be tuned for controlling the antagonistic gradient profiles and therefore the reaction conditions. Multiple channels may also be integrated parallelly into the gel and inject with reactants of different concentration, thereby enabling the authors to sweep through multiple parameter parallelly and find condition when the RD system can form desired spatiotemporal pattern. The ability to screen through large parameter space is demonstrated with the bromate-sulfite-ferrocyanide (BSF) chemical oscillator. By running different acidic condition parallelly, the authors found out the system only form spatio temporal oscillation in acidic condition and falls into a stationary stripe state instead when the acid concentration is not high enough.

Finally, the authors also mentioned that the rim and chemical composition of the gel are two more knobs that can be tuned to control the gradient profiles by changing the diffusion coefficient of the gel. While the change of chemical composition is not demonstrated with experimental result, probably due to the fact of the difficulty in quantifying the change in diffusion coefficient, the authors used simulation to show how the effect in the the change of the rim would form different spatiotemporal patterns. The authors constructed a RD model of the BSF reactions and found out the system falls into three different states under different geometric conditions, and, also produces two different waves. The result of the model provides good intuition of how the RD system should be tuned to achieve certain pattern and serves as a great map and guidance to how the RD system should be used and built.

Comments:

The authors mentioned that the variable composition of the hydrogel as an additional chemical control parameter. The authors can also run some simulation via their current existing model to provide proof and also how different diffusivity would affect the RD system, and therefore provide qualitative intuition of how the variable composition should be tuned when intended.

The final figures for the simulation result may be a bit confusing to the readers and require more explanation, as the system seems to be jumping back and from within states with no predictable pattern. For example, in fig 5(i) along the line of $w = 0.5$, the system jumps from state F to wave 1 to M1 and wave 1 again before entering state 2. The second wave 1 seems a bit weird and inconsistent and would probably makes more sense if it is a wave 2 instead, judging from the trend. Can the authors offers some explanation for such observation?

Answer for the comments of Reviewer #1:

“Actually, the design of the reactor should be the most instructive part of the paper shown in enough detail that the community can use it for various tasks. In this regard, I suggest the following three additions: (i) to provide a detailed description of the reactor design including the construction of the mold, types of the tubes, photos of different stages of the assembly process; (ii) to show designs with smaller channels (e.g. 1 mm in diameter) because the miniaturization of the setup will significantly broaden its applicability, especially to biochemical systems; (iii) to demonstrate a setup from polyacrylamide hydrogel because polyacrylamide hydrogels are much more elastic and less brittle than agarose hydrogels making them more applicable for designing functional materials. Finally, there are a few small issues that should be addressed: p.1, the term “communicating reaction zones” is unclear in the context of the abstract without reading the full text of the manuscript. A figure for the three-channel setup should be given. p.4 Regarding the following sentence “In both cases, the continuous supply of reagents maintains non-equilibrium conditions and stabilizes the three-zoned patterns”. It should be considered that in contrast to $[H^+]$ pattern, $CaCO_3$ pattern is not stationary because $CaCO_3$ crystals grow constantly.”

We have made a supplementary video which shows the reactor design and also each step of the preparation of the gel with flow-through channels. We hope that with the help of this movie one can easily reproduce our results.

Following the suggestion of the Reviewer we performed experiments in a miniaturized design with significantly smaller channels ($d=1.3$ mm) but with the same distance between the channels ($w = 2.5$ mm) and verified the existence of pH waves in the BSF reaction. These experiments are mentioned in the revised manuscript and presented in a supplementary figure.

We have also tested the use of polyacrylamide hydrogel. The main difficulty, in this case, is caused by the swelling of the gel. Therefore, after the polymerization, the gel was moved into a larger holder. The preparation of the polyacrylamide gel and the observed results are mentioned in the revised manuscript and presented in a supplementary figure.

We have modified slightly the abstract according to the suggestion of the Reviewer as “The use of additional channels can create interacting reaction zones.”

We have modified the criticized sentence as “In both cases, the continuous supply of the reagents provides non-equilibrium conditions and therefore maintains the three-zoned patterns.”

Answer for the comments of Reviewer #2:

“I have only minor comments: To be clear, in figure 2, the space shown is that between the two channels, w , and the crystals shown in 2d/e are formed in the gel matrix, not the liquid channel? Is it possible to take images from the side also, to view the 3 dimensional structure of the Turing patterns?”

In figure 2 the region between the two channels are shown. We did not observe precipitation in the channels.

The actual reactor design does not allow to take images from the side. We have made a few preliminary experiments in a reactor by using a thin gel sheet with separated channels, which can provide this information. Another possible solution would be to make a cylindrical gel with flow-through channels and apply tomography to really see the 3D structures. We have not tested it but very likely this would be the best solution of this problem.

Answer for the comments of Reviewer #3:

“Comments:

The authors mentioned that the variable composition of the hydrogel as an additional chemical control parameter. The authors can also run some simulation via their current existing model to provide proof and also how different diffusivity would affect the RD system, and therefore provide qualitative intuition of how the variable composition should be tuned when intended.

The final figures for the simulation result may be a bit confusing to the readers and require more explanation, as the system seems to be jumping back and forth within states with no predictable pattern. For example, in fig 5(i) along the line of $w = 0.5$, the system jumps from state F to wave 1 to M1 and wave 1 again before entering state 2. The second wave 1 seems a bit weird and inconsistent and would probably makes more sense if it is a wave 2 instead, judging from the trend. Can the authors offers some explanation for such observation?”

By following the suggestion of Reviewer #1, we have made experiments in polyacrylamide gel. Polyacrylamide gels always contain free carboxyl groups which affect the diffusivity of H^+ . In these experiments, we observed the formation of stationary Turing structures, due to the long range inhibition caused by the gel matrix. This is a demonstration of the use of the composition of the hydrogel as an additional chemical control parameter.

We thank the remark of the Reviewer in Figure 5. In the revised manuscript we point out that the transition between the two types of waves is smooth. We have checked carefully this point in the phase diagram and corrected it.

We thank again the Reviewers for their comments and suggestions and critical reading of our work. We hope that the changes we made will prove satisfactory. At the same time, we remain open to further suggestions as to how to improve the manuscript.

Sincerely yours,

Istvan Szalai
Professor of Chemistry

REVIEWERS' COMMENTS:

Reviewer #1 (Remarks to the Author):

The authors properly addressed my critics. The revised manuscript can be accepted as is.

Reviewer #3 (Remarks to the Author):

We are satisfied with the revision and response of the authors. Also, the addition of an introduction video has been incredibly helpful for readers to understand the setup of the system.